# Abattoir Wastewater Treatment in Anaerobic Co-Digestion with Sugar Press Mud in Batch Reactor for Improved Biogas Yield

**Beatrice N. Anyango** [1,2,*], **Simon M. Wandera** [2] **and James M. Raude** [3]

1   Department of Water, Environment and Natural Resources (DWEINR), County Government of Busia, Busia Private Bag-50400, Kenya
2   Department of Civil, Construction and Environmental Engineering, Jomo Kenyatta University of Agriculture and Technology, Nairobi P.O. Box 62000-00200, Kenya
3   Soil, Water and Environmental Engineering Department (SWEED), Jomo Kenyatta University of Agriculture and Technology, Nairobi P.O. Box 62000-00200, Kenya
*   Correspondence: bettyanyango1234@gmail.com; Tel.: +254-712-807-303

**Abstract:** Slaughterhouse wastewater (SHWW) has a great potential to generate biomethane energy when subjected to anaerobic digestion (AD). Nonetheless, the process is susceptible and prone to failure because of slow hydrolysis and the production of inhibitory compounds. Accordingly, to address this deficiency, anaerobic co-digestion (ACoD) is used to improve the treatment efficiency of the monodigestion of this high-strength waste and thereby increase methane production. The current investigation utilized the biochemical methane potential (BMP) test to assess the treatment performance of co-digested SHWW with sugar press mud (SPM) for improving biomethane energy recovery. It was established that the ACoD of SHWW with SPM increased methane ($CH_4$) yield, enhanced organic matter removal efficiency and improved process stability, while also presenting synergistic effects. The anaerobic monodigestion (AMoD) of SHWW (100SHWW: 0SPM) showed a higher $CH_4$ yield (348.40 $CH_4$/g VS) compared with SPM (198.2 mL $CH_4$/g VS). The 80% SPM: 20% SHWW mix ratio showed the optimum results with regard to organic matter removal efficiency (67%) and $CH_4$ yield (478.40 mL $CH_4$/g VS), with increments of 27% and 59% compared with AMoD of SHWW and SPM, respectively. However, it is also possible to achieve 5% and 46% $CH_4$ yield increases under a 40% SPM: 60% SHWW mix proportion in comparison to the AMoD of SHWW and SPM, respectively. Furthermore, kinetic analysis of the study using a modified Gompertz model revealed that the $CH_4$ production rate increased while the lag time decreased. The synergistic effects observed in this study demonstrate that incorporating SPM into the substrate ratios investigated can improve the AD of the SHWW. In fact, this represents the environmental and economic benefits of successfully implementing this alternative solution. Bioenergy recovery could also be used to supplement the country's energy supply. This would help to increase the use of cleaner energy sources in electricity generation and heating applications, reducing the greenhouse gas effect.

**Keywords:** anaerobic digestion; biochemical methane potential; co-digestion; kinetic model; mesophilic condition

## 1. Introduction

The meat sector in the agro-processing industry has received considerable critical attention as it contributes immensely to high-strength wastewater generation [1]. Agricultural water use accounts for 70% of the global pollution sources of nutrients and other contaminants, which, if left unmanaged, can result in significant social, economic, and environmental costs [2]. Meat production has more than quadrupled over the past 50 years. Each year, it is anticipated that more than 320 million metric tons of meat are produced globally [3]. Furthermore, according to Kenya Markets Trust [4], the annual per capita meat demand in emerging and developed nations is expected to rise from 25 to 37 kg and





88 to 100 kg, respectively, between 1997 and 2030. Moreover, beef demand in Kenya has increased, resulting in increased wastewater generation from slaughterhouses.

Meat processing plants that automate carcass dressing consume more water and produce effluent with a high content of protein and lipid-based organic matter. A key issue is the safe disposal of this wastewater, which is associated with increased risk of disease-causing microbes, representing a serious environmental hazard and a threat to human health [5]. Unfortunately, modern slaughterhouses pose a challenge to sustainably and adequately treating such organic waste. Therefore, SHWW is a growing public health concern worldwide. As a result, there is a need for a waste management strategy that is both economically and environmentally sound [6]. In that regard, conventional treatment methods (incineration and landfill) have high initial investment costs, with high energy demands and requiring highly skilled manpower. Therefore, the effective and adequate treatment of SHWW with sustainable and effective construction, including operation, is critical [7].

To date, several studies have recognized the contribution of anaerobic digestion (AD) to the treatment of organic wastes [8–12]. In contrast to traditional waste management procedures, AD produces bioenergy, while effluent provides substrate for recycling nutrients [13]. SHWW is of concern because of its high protein and lipid levels, hence being preferred for biomethanation. Previous studies have established the limitation of AMoD of SHWW, afflicted by the inhibition of functional microorganisms due to toxic free ammonia ($NH_3$) [8,14,15]. Moreover, the rapid hydrolysis of lipids results in the accumulation of volatile fatty acids (VFAs) which lower the bioreactor pH and adversely affect methanogens. As a result, the AMoD of SHWW on an industrial scale is limited [16]. Long et al. [17] also stated that the AD of wastes high in lipid content presents operational challenges. Some of the problems associated with lipid AD include methanogenic archaea inhibition; substrate and product transfer limitations; sludge flotation; digester foaming; and pipe obstructions. Moreover, an unbalanced C/N ratio and low buffering capacity make the digestion of SHWW as a single substrate problematic in practice [18].

Recently, researchers have shown an increased interest in ACoD [9,19,20]. Therefore, introducing co-substrates may be a viable solution to increase process stability and $CH_4$ generation from the AD of SHWW. However, these co-substrates ought to have excellent buffering capacity and be rich in carbon sources that can boost the C/N ratio [19]. As opposed to AMoD, ACoD delivers macro and micronutrients, enhanced buffer capacity, microbial consortia, process stability, dilute inhibitors, a balanced C/N ratio, and increased methane yield [11,21].

There is an increased focus on SPM, an energy-crop-rich agro-waste, as one of the most suitable co-substrates [10,22,23]. Unfortunately, SPM composts emit an obnoxious smell and toxic sulphur dioxide ($SO_2$) and sulphur trioxide ($SO_3$) gases upon burning their briquettes, which pollute the environment. Therefore, there is an urgent need to address the safety problems caused by SPM. ACoD is important because it helps with proper agro-waste management and improves treatment efficiency [19]. When the digestate is rich in beneficial plant nutrients such as nitrogen and organic carbon, the environment benefits [5].

Despite its suitability as a substrate in AD, SPM contains wax and a complex ligno-cellulose material that makes it difficult to hydrolyse, limiting its bioconversion efficiency and therefore the treatment of organic matter in AD systems [8,20]. Furthermore, the slow hydrolysis of cellulosic carbohydrates in the SPM explains its low bioavailability for bioconversion processes, which requires longer retention times and reduces the efficiency of bioconversion and hence the performance of the biogas plants [22,24]. Fortunately, substrate digestibility, system stability, and reactor performance can all be improved through strategies such as pre-treatment [25–28], two-stage AD systems [25,29]), pH control by chemical alkali [20,30], temperature selection [9,25] and ACoD [10,11].

To date, a large and growing body of literature has investigated the concept of ACoD. Rahman et al. [9] and Rouf et al. [22] observed methane yields of 254 and 241 mL $CH_4$/g

VS, respectively, from the AMoD of SPM. In contrast, Cárdenas-Cleves et al. [31] reported a relatively lower $CH_4$ yield for AMoD of SPM (67.67 mL $CH_4$/g VS). The discrepancy is traced back to sugar variety, soil quality, nutrients applied in the field, the clarification method utilized, and perhaps other environmental elements [20]. Moreover, the survey conducted by Rouf et al. [22] indicated that the ACoD of SPM with cane pith in a 1:1 mix ratio yielded 381 $CH_4$/g VS of $CH_4$, while the ACoD of SPM with cow dung in a 1:1 mix ratio yielded 167 $CH_4$/g VS of $CH_4$ in a batch reactor. The findings agree with those documented by Janke et al. [32] where a 50% increase in $CH_4$ production was observed under the semi-continuous ACoD of SPM with bagasse. Additionally, an improvement in $CH_4$ yield, from 171 mL $CH_4$/g VS to 249 mL $CH_4$/g VS, was observed by Ma et al. [33] for co-digesting animal manure (cattle, poultry, and swine manure) with other feedstocks (crop residues, food waste, and micro-algae). Furthermore, Mozhiarasi et al. [34] explored the "influence of pre-treatments and ACoD of slaughterhouse waste with vegetable, fruit, and flower market wastes (VMW) for enhanced $CH_4$ production". ACoD improved process stability and led to a $CH_4$ yield of 273.2 mL $CH_4$/g VS in a 1SHW:3VMW mix ratio. Similarly, Salehiyoun et al. [5] reported more than an approximate 50% increment in the ACoD of waste mixed sludge and slaughterhouse waste. Elsewhere, Pagés-Díaz et al. [35] discovered that there was no inhibition in the ACoD of slaughterhouse waste with animal manure. In order to make comparisons with other investigations, to the best of our knowledge, nobody has explored the ACoD of SHWW with SPM. Nonetheless, our findings are consistent with those of other studies of a similar nature, as summarized in Table 1.

**Table 1.** Performance of the ACoD of slaughterhouse wastes with different substrates.

| Reference | Co-Digested Feedstock | Operation Conditions | Improvements |
|---|---|---|---|
| [5] | SHWW with WMS | Batch and CSTR; mesophilic temp; HRT 18d, 13.5d, 11d; OLR 1.5 kg VS/m³ d | 50% methane increase |
| [36] | Poultry SHWW with sewage sludge | Batch mode; mesophilic temp; HRT 50d, 42d | 63% VS removal; 88% COD reduction |
| [9] | Poultry droppings and SPM | Batch and CSTR; mesophilic and thermophilic temp; HRT 20d | Methane increased by 8 and 29% in contrast to AMoD of PM and PD, respectively |
| [37] | OMW with SHWW | batch and continuous ASBR; mesophilic temp; OLR 10 g COD/L/day; HRT 20d | Reactor degraded 10 g COD/L/day |
| [38] | SHWW with rendering plant | CSTR; mesophilic and thermophilic temp; 1.0 and 1.5 kg VS/m³ day OLRs; 50 d HRT | 262–572 mL $CH_4$/g VS added |
| [39] | SHWW with OFI (75% SHWW: 25% OFI) | semi continuous; mesophilic temp; OLR 64 g VS $L^{-1}$ $day^{-1}$ | 57% (*v/v*) methane increase |
| [40] | FVW and AWW (30%FVW:70%AW) | Single-stage ASBR; mesophilic temp; 20 d, 10d HRT; 2.56 g VS $l^{-1}$ $day^{-1}$ OLR | 75% more methane yield |

**Table 1.** *Cont.*

| Reference | Co-Digested Feedstock | Operation Conditions | Improvements |
|---|---|---|---|
| [41] | AWW with PPWW and/or RPS | Batch; mesophilic temp; HRT 22 d | 50 PPWW:50 pig slurry achieved 72% VS removal and 35 mL average daily methane production; 32% max methane |
|  |  |  | No improvement for AWW due to poor buffering and low pH |
| [42] | Cattle AWW with FVW (50%AWW:50%FVW) | Unstirred two-staged ASBR; mesophilic temp; semi-continuous fed | 70.26% more methane yield; 57.11% VS reduction compared to AMoD of AWW |

SHWW—slaughterhouse wastewater; SPM—sugar press mud; AWW—abattoir wastewater; FVW—fruit and vegetable waste; WMS—waste mixed sludge; OMW—olive mill wastewaters; PM—poultry droppings; SHWs—slaughterhouse wastes; OFI—Opuntia fícus-indica; PPWW—potato processing wastewater; RPS—raw pig slurry; CSTR—continuously stirred tank reactor; OLR—organic loading rate; HRT—hydraulic retention time; VS—volatile solids; ASBR—anaerobic sequencing batch reactors.

The BMP of the substrate and organic matter removal efficiency could provide a comprehensive judgment of the performance of an AD bioreactor for treating organic waste. Previous studies agree that treating agro-processing organic wastes in AD highlights the ability for organic degradation and $CH_4$ yield as a source of biofuel [25,26]. Because SPM is a good carbon source, it is employed to help stabilize the fermentation of SHWW, which is high in protein. Whilst extensive research has been carried out on the ACoD of agro-wastes, no single study exists which has directly co-digested SHWW with SPM in AD, and very little is known about the optimum mix proportion of these two substrates. The unique feature of this project is the use of SPM as a co-substrate in the AD of nitrogen-rich organic wastes to optimize process performance. This paper analyses the impact of different mixing ratios in the ACoD of SHWW and SPM on $CH_4$ yield and organic decomposition. There are three primary aims of this study: (i) to determine the $CH_4$ yield of SHWW co-digested with SPM at different mix ratios; (ii) to determine the impacts of co-digestion on treatment efficiency; and (iii) to establish the organic degradation kinetics using a modified Gompertz model.

## 2. Materials and Methods

### 2.1. Seed Sludge and Substrates

The SHWW samples were collected from a cattle abattoir in Juja's outskirts, in Kiambu County, Kenya, while SPM samples were collected from Busia Sugar Industry (BSI), in Busia County, Kenya. Fresh, raw SPM samples gathered from the factory were used at 6% TS. Fresh inoculum samples were collected from a biogas digester treating dairy manure using 2-liter containers and preserved in a temperature-controlled incubator to maintain the temperatures at 37.0 ± 1.0 °C. The samples of SHWW and SPM were labelled, sealed, and refrigerated at 4 °C before conducting analyses and testing to minimize undesirable fermentation processes. SHWW was not diluted or modified before or after storage. The initial TS and VS content of the substrates and the inoculum are presented in Table 2.

**Table 2.** Characteristics of substrates and inoculum.

| Parameter | SHWW | SPM | Inoculum |
|---|---|---|---|
| TS (%) | 3.5 | 6.3 | 7.1 |
| VS (%) | 3.2 | 5.7 | 6.3 |

The TS and VS for the raw substrates and inoculum before dilution.

## 2.2. Batch Experiment Set up and Operation

One of the most well-known methods for assessing the $CH_4$ potential and biodegradability of wastewater and waste biomass is the BMP test [43]. In the current study, BMP tests were prepared according to the procedure used by Wandera et al. [44]. The batch assays were conducted using 125 mL glass digesters with a working volume of 80 mL sealed with rubber stoppers. The blending ratios between SPM and SHWW and the substrate to inoculum ratio on a VS basis used in this study are presented in Table 3. SPM was sieved to a particle size of 0.42 mm and dissolved with tap water to achieve the target total solid (TS) concentration of 6% on a wet weight basis and then fed into the digesters after fully mixing. Control and each mix ratio were conducted in triplicate. Before commencement, all reactors were purged with nitrogen gas for around 5 min to remove air from the headspace and to help ensure an anaerobic environment. These lab-scale digesters were subsequently placed into a lab incubator (Model SV-05E/09E/23E, Lab Companion, Isuzu Seisakusho Co., Ltd., Sanjo, Japan) at a mesophilic temperature of $37.0 \pm 1.0\,°C$. The digesters were shaken manually twice every day for about 1–2 min.

**Table 3.** The mixing ratio of the substrates.

|            | Inoculum (mL) | SPM (mL) | SHWW (mL) |
|------------|---------------|----------|-----------|
| Control    | 80            | 0        | 0         |
| MIX 00:100 | 40            | 0        | 40        |
| Mix 100:00 | 40            | 40       | 0         |
| Mix 80:20  | 40            | 32       | 8         |
| Mix 20:80  | 40            | 8        | 32        |
| Mix 60:40  | 40            | 24       | 16        |
| MIX 40:60  | 40            | 16       | 24        |
| MIX 50:50  | 40            | 20       | 20        |

## 2.3. Kinetic Modelling

Several authors have modelled the batch experiment data and have been found to produce reasonable predictions of full-scale behaviour [44,45]. Kinetic modelling is widely used in predicting $CH_4$ yields, establishing key parameters for reactor design and optimizing the performance of AD processes. For simulating $CH_4$ accumulation data with a lag phase ($\lambda$), which is a critical factor in determining AD efficiency, the modified Gompertz model is preferable [46]. In addition, where toxic substances or bio-resilient organic compounds exist, this kinetic model is more suitable [47]. A modified Gompertz model, expressed by Equation (1), was integrated into the model to simulate the batch experimental data [48].

$$F_{(t)} = F_o \cdot \exp\left\{ -\exp\left[ \frac{R_{max} \cdot e}{F_o}(\lambda - t) \right] + 1 \right\} \tag{1}$$

where $F_{(t)}$ is the cumulative methane yield at digestion time t in mL $CH_4$/g-$VS_{added}$; $F_0$ is the substrate's methane potential in mL $CH_4$/g-$VS_{added}$; $R_{max}$ is the maximum methane production rate in mL $CH_4$/g-VS.d; and e = 2.718281828.

The predicted $CH_4$ yield and the constants $F_0$, $R_{max}$, and $\lambda$ were determined by a non-linear least-square regression analysis conducted in the SPSS program (IBM SPSS Statistics 17 (2008)). The predicted $CH_4$ yield was plotted against the measured $CH_4$ yield using MS Excel 2013.

### 2.4. Statistical Evaluation

To evaluate if the model prediction fits with the experimental data, a statistical indicator, namely root mean square error (RMSE), was determined based on Equation (2) [45].

$$\text{RMSE} = \left( \frac{1}{n} \sum_{m=1}^{n} \left( \frac{d_m}{Y_m} \right)^2 \right)^{\frac{1}{2}} \tag{2}$$

where n is the number of data points; m is the mth measurements; Y is the measured $CH_4$ yield (mL $CH_4$/g VS); and d is the difference between the experimental and predicted $CH_4$ yield.

Furthermore, the statistical validity of the acquired cumulative methane yields from various mix ratios was calculated using an ANOVA (Analysis of Variance) test followed by Tukey's test in Microsoft Office Excel with a *p*-value of 0.05 as the limit.

### 2.5. Analytical Techniques

The daily produced $CH_4$ volume from each reactor was measured with a gas-tight syringe and then converted to the volume under standard temperature and pressure (STP, 0 °C and 101 kPa). The methane content was analysed for methane ($CH_4$, %) and carbon dioxide ($CO_2$, %) using gas chromatography (GC 7890 A, Agilent, Santa Clara, CA 95051, USA) fitted with a thermal conductivity detector and a stainless-steel column (13803-U, Sigma-Aldrich, Saint Louis, MO, USA). The splitless inlet, oven, and TCD detector temperatures were all kept at 60, 70, and 200 °C, respectively. The $CH_4$ and $CO_2$ were measured by a dual-wavelength infrared cell with a reference channel. The certified gases $CH_4$ (60, 15.01%) and $CO_2$ (40, 15.01%) were used to calibrate the gas analyser. Argon gas was used as the carrier gas in the GC, while nitrogen was used as the makeup gas. The GC was calibrated using standard gases consisting of $CH_4$ (60%) and $CO_2$ (40%) on a volume basis (*v/v*).

TS and volatile solids (VS) were determined in triplicate in homogenously mixed samples using standard methods [49]. The VS reduction was accounted for on the basis of microorganism growth. For TCOD and SCOD analysis, the closed reflux technique was used. The pH readings were taken from the samples directly via a portable pH meter (pH3210, Germany). The Nessler method was used to measure ammonium nitrogen ($NH_4^+$-N) concentration and was determined using a Shimadzu UV-VIS-1800 spectrophotometer (DR 2500, Hach, IA, USA). The elemental compositions of C, H, O, and N were determined utilizing an elemental analyser (AAS iCE 3300).

### 2.6. Analysis of the Substrate's Bioenergy Conversion Capacity

The theoretical methane potential (TMP) of the feedstock (SPM and SHWW) was estimated using Boyle's (modified Buswell) Equations (3) and (4) depending on the elemental composition [50,51]. The prediction of TMP was founded on the following assumptions: the conditions for microbial and substrate digestion are ideal; mixing was completed; and the temperature was maintained at a constant. The elemental analysis was undertaken to determine the percentages of the following key elements: C, H, O, and N. The measured output from the batch reactor was biogas, the composition ($CH_4$ and $CO_2$) of which was tested.

$$\begin{aligned}
C_nH_aO_bN_c + &\left( n - \frac{a}{4} - \frac{b}{4} + \frac{3c}{4} \right) H_2O \\
\rightarrow &\left( \frac{n}{2} + \frac{a}{8} - \frac{b}{4} - \frac{3c}{8} \right) CH_4 + \left( \frac{n}{2} - \frac{b}{4} + \frac{3c}{8} \right) CO_2 \\
&+ cNH_3
\end{aligned} \tag{3}$$

$$\text{TMP}\left( \frac{\text{mL } CH_4}{\text{gVS}} \right) = \frac{22.4 \times 1000 \times \left( \frac{n}{2} + \frac{a}{8} - \frac{b}{4} - \frac{3c}{8} \right)}{12n + a + 16b + 14c} \tag{4}$$

where $C_aH_bO_cN_d$ is the chemical formula for the substrates derived experimentally; a, b, c, d are the atomic masses of carbon (12), hydrogen (1), oxygen (16), and nitrogen (14),

respectively; and TMP is the theoretical methane potential at standard temperature and pressure (STP).

Equation (5) was also used to calculate the synergistic effect of ACoD of SHWW with SPM [47].

$$\text{Synergistic Increase in CH}_4 \text{ yield}$$
$$= \frac{\text{BMP}_{\text{co-digestion}}}{\text{BMP}_{\text{SPM}} \times \%\text{SPM} + \text{BMP}_{\text{SHWW}} \times \%\text{SHWW}} \quad (5)$$

where $\text{BMP}_{\text{co-digestion}}$ is the biomethane potential of the co-digestion sample (mL $CH_4$/g VS); $\text{BMP}_{\text{SPM}}$ is the experimental biomethane potential measured in the AD of SPM (SPM: SHWW 100: 0 ratio) (mL $CH_4$/g VS); %SPM is the percentage of SPM in the ratio; $\text{BMP}_{\text{SHWW}}$ is the experimental biomethane potential recorded in the AD of SHWW (SPM: SHWW 0:100 ratio) (mL $CH_4$/g VS); and %SHWW is the percentage of SHWW in the ratio.

When the increase in $CH_4$ yield is >1, a synergistic effect (S) occurs; =1, no synergistic effect (N); <1, the effect is antagonistic (A) [47].

Equation (6) was also used to calculate the methane yield (mL $CH_4$/g $VS_{\text{added}}$) [52].

$$\text{Methane yield} = \frac{\text{BMP}_{\text{cum}}}{\text{TS}_{\text{added}}} \quad (6)$$

where $\text{BMP}_{\text{cum}}$ is the cumulative methane yield (mL) and TS $_{\text{added}}$ is the weight (g) of total volatile solids fed to the digester. Additionally, the removal efficiencies of COD, VS, and TS were calculated using Equation (7).

$$\text{Removal efficiency}(\%) = \frac{G_i - G_f}{G_i} \quad (7)$$

where $G_i$ and $G_f$ represent the initial and final concentrations of the parameters, respectively.

## 3. Results and Discussion

### 3.1. Substrates Characteristics

Table 4 presents the summarized characteristics of substrates and fresh sludge. The VS/TS% of SHWW and SPM was 90 and 91%, respectively, which was found to be more suitable for AD, as previously reported by Jeung et al. [53]. Additionally, SHWW showed a high organic matter content as measured by COD (16 g $L^{-1}$), TS (3.5%) and VS (3.2%). Ultimately, the organic matter content of SHWW was remarkably higher than previously observed by Hernández-Fydrych et al. [54] and Bouallagui et al. [40], possibly due to solid separation prior to sampling and the fact that SHWW composition varies with sacrificed meat type, daily rate of processing, and butchering operational processes (washing, cutting, etc.) [5]. The sampled SHWW, in particular, contained blood, which resulted in a high COD concentration. However, the organic matter content was enriched when SHWW was mixed in different proportions with SPM diluted to 6% TS (Table 4). Therefore, combining substrates makes a good feedstock for AD experiments.

The C/N ratio is a crucial element for AD. The low C/N ratio (9.65) exhibited in SHWW is similar to prior findings [18]. According to Bouallagui et al. [40], low C/N ratios emerge if nitrogen is in excess, resulting in $NH_3$ accumulation and, as a result, pH values that choke the $CH_4$-producing bacteria. On the contrary, a high C/N ratio means that methanogens are quickly utilizing nitrogen in a way that results in a deficiency in the AD process, resulting in a drop in $CH_4$ yield. Therefore, a stable C/N ratio must be maintained to maximize $CH_4$ generation and process stability. The best C/N ratio, according to the literature, is about 20 to 30 [20]. This is efficient for microbial metabolic activities and adequate to sustain system operation and satisfy nutrient and energy needs for cell growth [55]. Consequently, substrates rich in C/N ratios have a low buffering capacity and are associated with high VFA accumulation during digestion. Substrates low in C/N ratios, on the other hand, have a high buffering capacity potential. As a result, elevated $NH_3$ concentrations during the digestion process limit microorganism growth.

Consequently, SHWW is a problematic substrate for biogas plants due to perceived $NH_3$ inhibition and an unbalanced C/N ratio [18].

**Table 4.** Characteristics of SHWW, SPM and inoculum with respective standard deviation (SD).

| Parameter | SPM 100:00 | SHWW 00:100 | Inoculum (Control) | MIX 80:20 | MIX 20:80 | MIX 60:40 | MIX 40:60 | MIX 50:50 |
|---|---|---|---|---|---|---|---|---|
| | Mean (±) SD | Mean (±) SD | Mean (±) SD | Mean (±) SD | Mean (±) SD | Mean (±) SD | Mean (±) SD | Mean (±) SD |
| TS (%) | 6.3 ± 0.3 (5.1 ± 0.3) | 3.5 ± 0.3 (2.4 ± 0.3) | 7.1 ± 0.3 (3.1 ± 0.3) | 6.2 ± 1.5 (4.5 ± 1.5) | 6.1 ± 0.4 (5.7 ± 0.4) | 5.98 ± 0.2 (5.8 ± 0.2) | 5.89 ± 0.8 (5.6 ± 0.8) | 6.3 ± 0.6 (5.2 ± 0.6) |
| VS (%) | 5.7 ± 0.6 (2.5 ± 0.6) | 3.2 ± 0.3 (1.5 ± 0.3) | 6.3 ± 0.3 (2.4 ± 0.3) | 4.3 ± 0.8 (1.4 ± 0.8) | 5.0 ± 0.3 (2.5 ± 0.3) | 4.2 ± 1.1 (2.4 ± 1.1) | 4.9 ± 0.6 (2.0 ± 0.6) | 3.8 ± 0.5 (1.8 ± 0.5) |
| VS/TS (%) | 90 | 91 | 90 | 70 | 82 | 70 | 83 | 60 |
| VS removal (%) | 60 | 53 | 62 | 67 | 50 | 43 | 59 | 52 |
| pH | 5.41 ± 0 (7.76 ± 0) | 8.06 ± 0 (8.34 ± 0) | 7.2 ± 0 (7.69 ± 0) | 7.2 ± 0 (8.10 ± 0) | 7.3 ± 0 (8.25 ± 0) | 7.1 ± 0 (8.21 ± 0) | 7.2 ± 0 (8.23 ± 0) | 7.3 ± 0 (8.20 ± 0) |
| TCOD (g $L^{-1}$) | 7.36 ± 0 (5.16 ± 9) | 16 ± 0.1 (12 ± 6.1) | 15.0 ± 0.1 (11.0 ± 9.1) | 10.8 ± 0.1 (8.3 ± 6.1) | 12.2 ± 0.3 (9.5 ± 9.3) | 11.8 ± 0.1 (8.3 ± 8.1) | 14.6 ± 0 (11.4 ± 7) | 15.6 ± 0 (12.7 ± 4) |
| COD removal (%) | 30 | 25 | 27 | 23 | 22 | 30 | 22 | 16 |
| $NH_4^+$-N (mg $L^{-1}$) | 1300 ± 3.3 (1205 ± 0.3) | 6407 ± 5.5 (4208 ± 0.5) | 1097 ± 8.7 (674 ± 0.7) | 5521 ± 7.3 (2426 ± 0.3) | 6887 ± 9.7 (2151 ± 0.7) | 6463 ± 4.6 (3841 ± 0.6) | 6323 ± 8.3 (4560 ± 0.3) | 6671 ± 6.5 (4500 ± 0.5) |
| C (%) | 27.28 ± 0.2 | 32.62 ± 0.1 | / | / | / | / | / | / |
| H (%) | 16.51 ± 0.6 | 17.88 ± 0.7 | / | / | / | / | / | / |
| O (%) | 1.37 ± 0.7 | 2.45 ± 0.4 | / | / | / | / | / | / |
| N (%) | 1.04 ± 0.4 | 3.38 ± 0.3 | / | / | / | / | / | / |
| C/N ratio | 26.23 | 9.65 | / | / | / | / | / | / |

Notes: n = 3; the first values refer to the substrates and mixtures before AD; the second values in brackets refer to the respective digestate after AD.

pH is a vital factor in AD. Therefore, the ideal pH for methanogenesis ranges from 6.5 to 8.2 [56]. In that regard, the current investigation recorded pH values of 8.06 and 5.41 for SHWW and SPM, respectively. However, the activity of methanogenic microorganisms involved in the digestion process decreases at a higher or lower pH. This emphasizes the importance of ACoD in optimizing reactor buffer capacity and AD efficiency. The pH level for SHWW remained above 8.0 for almost the entire process due to the relatively higher $NH_4^+$-N concentration (6407 mg/L) (Table 4), which was caused by the degradation of the proteins in SHWW. Bayr et al. [38] also reported similar scenarios during the AD of rendering plant and slaughterhouse waste under mesophilic and thermophilic conditions. These findings imply the possibility of $NH_3$ inhibition during the AD of the explored substrates, potentially inhibiting methanogenesis.

In contrast, the SPM reactor had a low initial pH because of its acidic nature. Consequently, it did not recover fully despite having a C/N ratio (26.23) well within the allowable threshold. This implies that the buffering capacity of the system was insufficient to keep a pH level within the satisfactory limits for AD. Moreover, the C/N ratio of SPM (26.23) is in good agreement with that reported in previous studies [20,21]. Previous studies have reported that ACoD improved the C/N ratio of the digestate and minimized the toxicity of nitrogen in the form of $NH_3$ [8,57]. However, the lignocellulose nature of SPM limits its biodegradability, therefore hindering the AD process. During the anaerobic co-digestion of SPM with food, Cárdenas-Cleves et al. [31] discovered similar results. According to Qamar et al. [30], pH adjustment through alkali treatment can maintain the stability of the process. As for the 80% SPM: 20% SHWW mix ratio, the initial pH significantly increased from 7.2 to 8.10 in the effluent due to the consumption of VFAs, thus indicating the presence of a buffer effect that maintained optimal AD conditions.

Total and soluble COD, TS, and VS values showed the presence of a high content of organic matter in the studied substrates (Table 4). According to Monou et al. [41], VS biodegradability strongly depends on original TS concentrations. With 53% VS destruction for SHWW, an 80:20 SPM: SHWW mix ratio yielded a 67% VS breakdown. The ACoD of SHWW with SPM was believed to have enhanced the biodegradability of the mixed medium. For instance, the large concentration of VS (57%) in the SPM as a co-substrate led to increased VS degradation. This could be credited to the digestion medium's synergistic effect and improved digestibility. However, a high VS breakdown in conjunction with a low maximum $CH_4$ content, on the other hand, denotes an imbalance between acidogenesis and methanogenesis. This is caused by low pH and nutrient deficiency [56].

During the trials, COD levels declined considerably due to the anaerobic breakdown process in all digesters. Overall, COD degradation was not attractive even for an 80% SPM: 20% SHWW reactor. One such scenario may imply that supplementary treatment is necessary immediately after AD so that the effluent can be unloaded into the surroundings in compliance with the applicable standards. Therefore, AD is regarded as a practical treatment method for SHWW due to its high COD and pathogen removal efficiency. Moreover, odour issues associated with abattoir effluents are limited [39].

### 3.2. Effect of ACoD on Biomethane Production

Table 5 presents the total $CH_4$ yield over the course of the study. The present study demonstrated that the $CH_4$ yield of the SPM80%:20%SHWW mix ratio was much greater than the AMoD of SHWW and SPM, respectively, even though the difference for the other mix ratios was relatively small. In fact, the $CH_4$ yield of SPM80%:20%SHWW (478.40 mL $CH_4$/g VS) was about 27% and 59% higher than in SHWW and SPM, respectively (Table 5). The mixing ratio influences the characteristics of each substrate; hence, improved $CH_4$ yield could be obtained through the self-buffering of the digestion medium [8]. Generally, SPM allowed for nutrient balance, thus balancing the C/N ratio in the process. This highlights the effectiveness of co-digesting SHWW with a carbon-rich waste such as SPM. Furthermore, ACoD allowed for increased organic matter content, implying that SHWW can be combined in such fractions with SPM. As a result, this would eventually impact the environment positively, leading to a cost reduction during biogas plant operation. Bohutskyi et al. [47] and Reyes et al. [8] discovered comparable results. These current findings are consistent with those of other studies of a similar nature [9,20].

**Table 5.** BMP of long-term mesophilic batch reactors with respective standard deviation (SD).

| | Control | SPM 100:00 | SHWW 00:100 | MIX 80:20 | MIX 20:80 | MIX 60:40 | MIX 40:60 | SPM: SHWW 50:50 |
|---|---|---|---|---|---|---|---|---|
| Methane yield (mLCH$_4$/VS) | 52.25 ± 72.3 | 198.2 ± 298.7 | 348.4 ± 285.2 | 478.4 ± 499.2 | 325.8 ± 404.7 | 302.1 ± 328.7 | 366 ± 400.8 | 357.8 ± 359.7 |
| Increase in methane yield | / | / | / | 2.10 | 1.02 | 1.17 | 1.27 | 1.31 |
| CH$_4$ content (%) | | 22 | 18 | 65 | 24 | 52 | 32 | 40 |
| TMP (mL CH$_4$/g VS) | / | 333.14 | 541.75 | / | / | / | / | / |

Notes: n = 66.

In relation to cumulative specific $CH_4$ yield, the bioreactors analysed were in the following sequence: 80%SPM: 20%SHWW > 40%SPM: 60%SHWW > 50% SPM: 50%SHWW > 100%SHWW > 20% SPM: 80%SHWW > 60% SPM: 40%SHWW > SPM > control. However, closer inspection of the ANOVA test shows that mixtures of 60%SPM: 40%SHWW, 50%SPM: 50%SHWW, and 20%SPM: 80%SHWW had an insignificant effect on $CH_4$ production ($p$ =1.000–0.884). This suggests that co-digesting two or more substrates is not a guarantee to achieve higher $CH_4$ yields than the AMoD of substrates in the mix. In fact, the AD process in SPM20%:80%SHWW, SPM60%:40%SHWW, and SPM biodigesters was inefficient.

Very little performance efficiency was attained in terms of organic matter degradability and methane production.

In this regard, the high $NH_4^+$-N concentrations (6887 mg $L^{-1}$) at the start of the experiment in 20% SPM:80% SHWW probably hindered the methanogens within the reactor, resulting in low methane yield. Furthermore, protein degradation in SHWW is depicted by relatively high $NH_4^+$-N concentrations (6407 mg $l^{-1}$), which imply the possibility of $NH_3$ inhibition during AD. However, SPM has the lowest $NH_4^+$-N concentrations (1300 mg $l^{-1}$) and is hence a suitable co-substrate. Indeed, Yenigün and Demirel [15] observed that high concentrations of free $NH_3$ above 55 mg $L^{-1}$ diffuse across cell membranes, causing microbial destruction and disrupting the entire AD process. This $NH_3$ toxicity could be mitigated through the ACoD of SHWW with agricultural wastes high in carbon, such as SPM. Furthermore, a good substrate mix stabilizes the C/N ratio and optimizes the mixture's buffering capacity, resulting in increased reactor efficiency [8].

In addition, mixing the substrates in 40%SPM: 60%SHWW showed a notable change in $CH_4$ yield ($p = 0.001$). Our findings suggest the existence of an ideal ratio of SHWW: SPM at which the ACoD of these feedstocks is simple. However, when the proportion was set at 80%SPM: 20%SHWW, there was an enormous improvement in the methane yield ($p = 0.000$). Hence, 80% SPM: 20% SHWW is the ratio for optimal methane yield during the ACoD of SPM and SHWW.

Furthermore, as shown in Table 5 and Figure 1, the AD of SHWW yielded a net $CH_4$ production of 348.40 mL $CH_4$/g VS, which is significantly higher than that of SPM. This confirms that the bacterial activities in SHWW digestion favour the entire fermentation process. This high $CH_4$ yield may also be due to the high protein and lipid content of SHWW, which is readily available [58–60]. Even so, a negligible amount of $CH_4$ was detected.

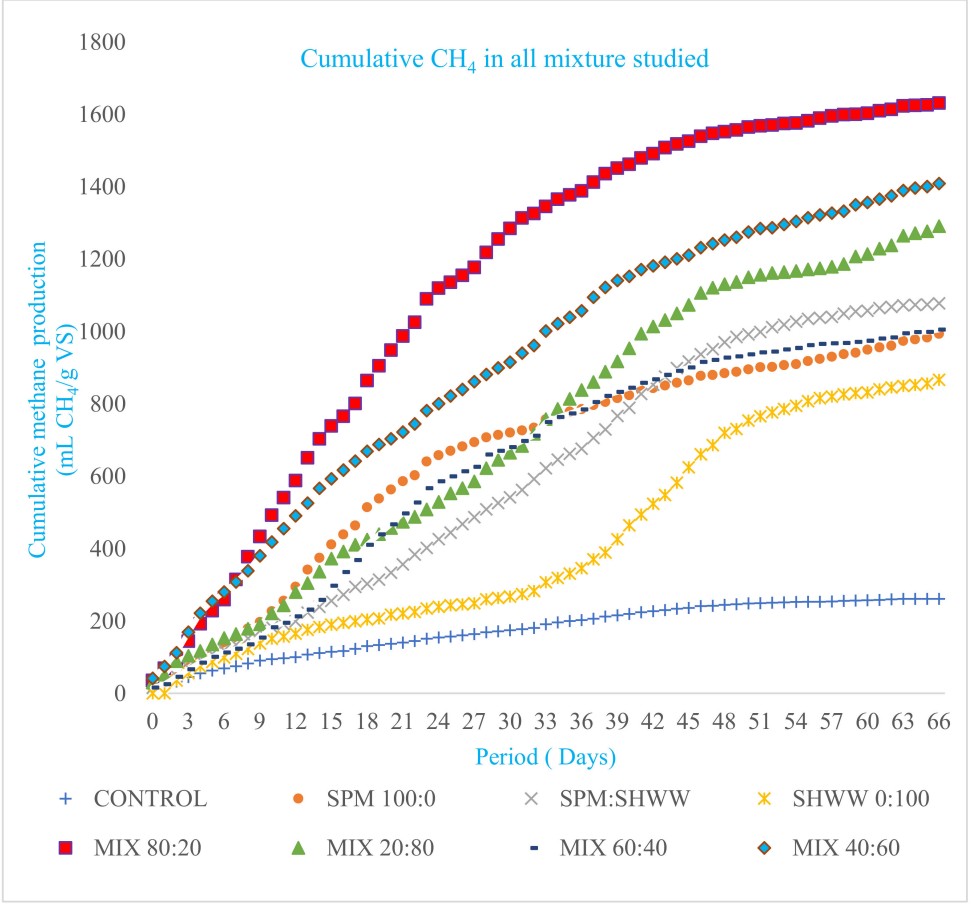

**Figure 1.** Cumulative methane yield in the investigated mix proportions.

SPM resulted in a low methane yield because of its high lignin content. It is possible that not all of the carbon in SPM is bioavailable for microbial degradation. As a consequence, the ACoD of SHWW with SPM stimulates methanogenic activity. This, however, comes at the expense of the digestion of SPM. Fortunately, the huge difference in the C/N ratios can be brought into the ideal range by blending SHWW and SPM in varying proportions. In addition, the low BMP presented in SPM agrees with the findings from other similar studies of low yields from lignocellulosic agro-wastes [8,20]. This is due to the slow hydrolysis of complex carbohydrates in lignocellulosic biomass, which requires long contact times to be hydrolysed by hydrolytic microbes. Furthermore, Talha et al. [20] observed that SPM's low $CH_4$ yield was due to its high ash content, probably related to cane variety with varying lignin content, soil conditions, and other environmental factors.

In addition, previous findings regarding SPM digestion point to the presence of process inhibitor factors [56]. Indeed, feedstock with a high C/N ratio has a low buffering capacity and generates an excess of VFAs during digestion, resulting in a pH drop [18]. At the start of acidogenesis, pH is most likely to drop before actually rising as the VFAs degrade to produce methane. In that regard, the accumulation of VFAs can probably cause process instability as it can reflect a kinetic uncoupling between the acidogenesis and acetogenesis phases of the AD process [56]. Herein, the inhibition reported in SPM is most inclined to this attribute.

Overall, current results also confirm that the presence of SPM impacted the $CH_4$ quality. SPM fermentation produced biogas with a low average $CH_4$ content (22%) when compared to the 80% SPM: 20% SHWW mix proportion. In fact, combining SPM80% with 20% SHWW resulted in a 65% increase in $CH_4$ content. However, an increase in SHWW portions (100%, 80%, and 60%) resulted in a decrease in average $CH_4$ content of only 18%, 24%, and 32%, respectively.

In particular, SHWW and SPM ACoD is perhaps the most effective approach. These observations are consistent with the results of [27], who assessed the energetic, economic, and environmental feasibility of the ACoD of pre-treated SPM. This analysis showed that both the environmental and energetic profiles as well as the profitability of $CH_4$ yield could be enhanced when the ACoD of SPM is considered. Abattoirs and sugar processing plants need to undertake mandatory waste management actions. In that regard, the ACoD of SHWW with SPM could potentially provide cost-effective alternatives.

Moreover, TMP recorded a higher maximum value from SHWW (541.75 mL $CH_4$/g-VS) compared with SPM (333.14 mL $CH_4$/g-VS) (Table 5). This can be explained by the more energy-rich lipids and proteins in SHWW than in SPM. In view of this finding, both substrates showed relatively low TMP. This could be due to the recalcitrance of SHWW and/or reduced methanogen activity because of an unbalanced C/N ratio [18,47]. With respect to SPM, low bioconversion is in line with the degradability of lignocellulosic biomasses [20,24]. Accordingly, this study agrees with other previous studies that agro-wastes pose digestibility difficulties during AD [14,15].

### 3.3. Kinetic Analysis

To describe the AD kinetics, the popularly utilized modified Gompertz model (Equation (1)) was fitted to specific $CH_4$ production in this study. This model links the rate of $CH_4$ production and microbial activity [61]. The λ of the $CH_4$ yield is accurately predicted by the modified Gompertz model. This is critical for modelling the AD of recalcitrant biodegradable feedstocks such as lignocellulosic materials [46,48]. The optimum proportion (80% SPM: 20% SHWW) exhibited the shortest λ, with the highest $R_{max}$, which was almost double compared to the AMoD of SPM and SHWW. This demonstrates that the 80%SPM: 20%SHWW mixture was readily biodegradable and contained functional microbial consortia that were more efficient for AD.

The λ for the AMoD of both SHWW and SPM was estimated to be relatively longer (6 d) compared to the mixed ratios. Thus, the high content of difficult biodegradable compounds could be the reason for the longer lag times in our study. The unbalanced C/N

ratio can also be blamed for the long λ in SHWW. In addition, the perceived inhibition due to free $NH_3$ and the rapid hydrolysis nature of SHWW reduce the rate of degradation [47]. Similarly, Cárdenas-Cleves et al. [31] reported an overall reduction in λ during the ACoD of SPM with food waste.

The simulation of the kinetic process also demonstrated that ACoD improved $R_{max}$ by between 77 and 90% and between 82 and 84% for the AMoD of SHWW and SPM, respectively. At the same time, the $R_{max}$ values for varying mix proportions were within a narrow range (Table 6). The implication of this finding is that all the mixed ratios had comparable effects on boosting methanogenic activity, despite the difference in their total optimum $CH_4$ yield. Nonetheless, our findings indicate that co-digestion helps to improve AD by boosting $R_{max}$ while significantly reducing λ. These improvements are critical for improving system economics because both variables ($R_{max}$ and λ) imply lower digester volumes and, as a result, lower capital investment and higher energy output from large-scale continuous AD systems [47].

The model simulated the experimental data with high accuracy and adaptability for both AMoD and ACoD, as presented in Table 6. The model's gradient shape accurately portrayed AD's lag, exponential, and stationary stages (Figure 2). Moreover, the simulated theoretical cumulative $CH_4$ data were consistent with the experimental values, implying that the suggested model represents an excellent fit. Accordingly, the co-efficient of determination ($R^2$) improved as the SPM percentage increased. $R^2$ was at its maximum value when the SPM was 80 and 60%, and was 0.995 when the SPM ratio was 50, 40, and 20%.

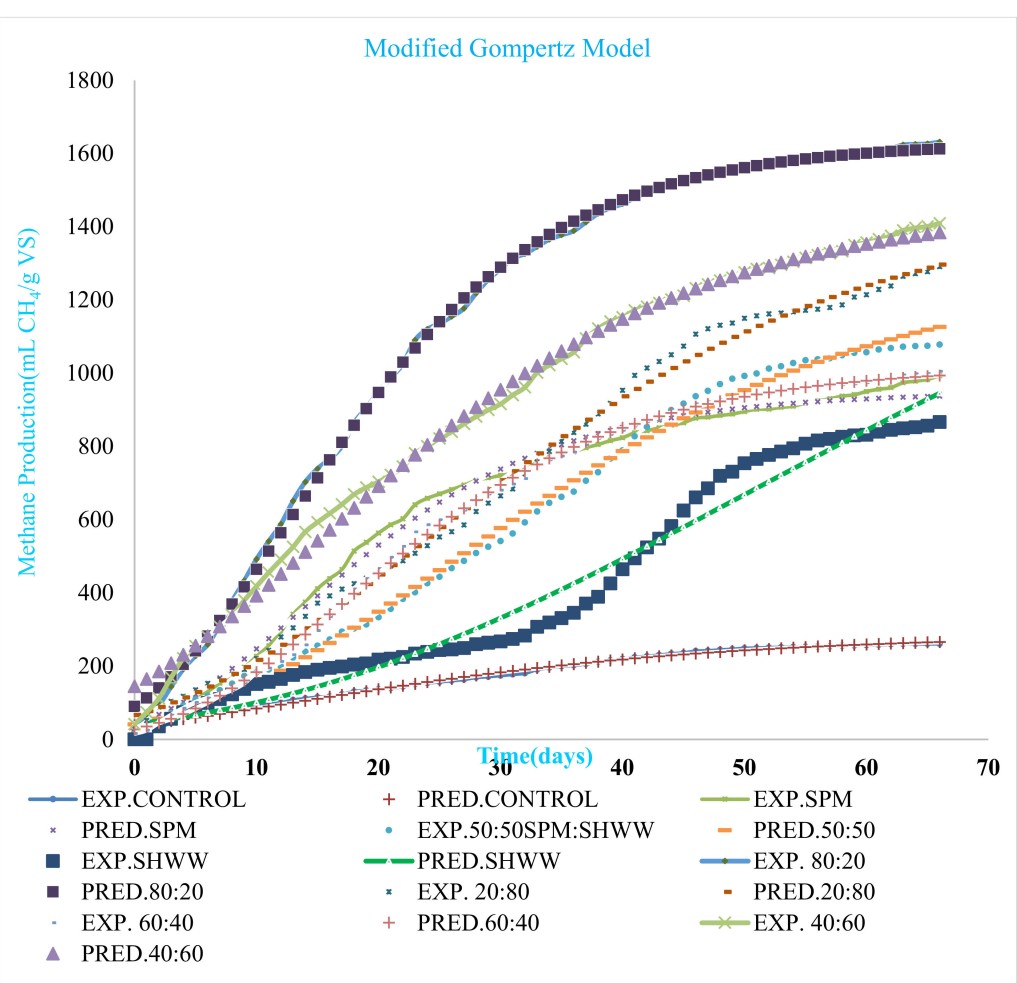

**Figure 2.** Experimental and simulation lines of cumulative methane production from different substrate mix proportions.

**Table 6.** Modified Gompertz kinetic modelling.

| Parameter | Control | SPM 100:00 | SPM: SHWW 50:50 | SHWW 00:100 | MIX 80:20 | MIX 20:80 | MIX 60:40 | MIX 40:60 |
|---|---|---|---|---|---|---|---|---|
| $P_o$ (mL $CH_4$/g VS) | 289.34 | 949.06 | 1299.57 | 887 | 1631.63 | 1478.96 | 1019.73 | 1464.39 |
| $R_{max}$ (mL $CH_4$/g VS/d) | 5.23 | 29.31 | 23.09 | 32.54 | 50.25 | 26.05 | 28.11 | 30.31 |
| $\lambda$(d) | 6.10 | 1.63 | 4.98 | 6.12 | 0.78 | 2.73 | 3.86 | 2.88 |
| $R^2$ | 0.993 | 0.994 | 0.995 | 0.995 | 0.999 | 0.995 | 0.999 | 0.995 |
| RSME | 5.75 | 23.05 | 25.42 | 49.44 | 11.55 | 27.43 | 17.58 | 28.48 |

Nevertheless, the $R^2$ values of the modified Gompertz model were high (0.993–0.999), which makes the model more adequate in our case. Similarly, Kafle and Chen [46] observed a lower $R^2$ value (0.994) for energy crops. Consequently, BMP is more significantly affected in plants' biomass since the cell membranes of their feedstock are shielded by a complex lignocellulosic material. In contrast to the present findings, Xu et al. [62] found a quite weak correlation ($R^2$ = 0.334) while using lignocellulosic biomass.

Additionally, the statistical indicator RMSE presented in Table 6 was used to evaluate whether the model prediction fits with the experimental data. The lower RMSE (11.55) and higher $R^2$ value (0.999) were calculated for the 80%SPM: 20%SHWW mixture followed by the 60%SPM: 40%SHWW mixture (RMSE = 11.54, $R^2$ = 0.999). Consequently, according to the kinetic analysis results (deviation between simulated and experimental methane yield and statistical indicators), the 80% SPM: 20% SHWW mixture was found to be the best mix proportion for the ACoD of SHWW and SPM.

## 4. Conclusions

The present study was designed to determine the impacts of co-digesting SHWW with SPM at different mix ratios on treatment efficiency and methane yield, and to evaluate the organic degradation kinetics using a modified Gompertz model. This study found that, generally, the organic matter removal and methane yield were improved while lag time was reduced upon co-digestion. The relevance of ACoD is evidently supported by the current findings. An implication of this is the possibility of the use of co-digestion to effectively remedy the possible ammonia inhibition that exists during SHWW fermentation and the inhibition induced by the readily acidifying SPM, which itself is naturally low in pH. The present study is suggested to be one of the first attempts to thoroughly examine the mesophilic ACoD of SHWW with SPM. However, the study is limited by the lack of information on the C/N ratio of different mix proportions. In spite of its limitations, the study certainly sheds new light on the rapidly expanding field of the ACoD of agro-wastes for the recovery of bioenergy, with the goal of the mitigation of CO2 emissions. A further study could assess the long-term effects of the composition and digestibility of the concerned agro-wastes in order to validate the consequences of their full-scale implementation. Repeating the study under a semi-continuous system to examine the stability of the AD process in practice would also be a fruitful area for further work. There is, therefore, a definite need for policies to be put in place to encourage the commercial production and distribution of biomethane, in order to encourage sustainable production and investment. However, unless the government creates incentives such as attractive fed-in tariffs, tax breaks, and public–private sector partnerships to encourage export-based methane generation, maximum methane production and the complete digestion of these agro-wastes will not be attained.

**Author Contributions:** Conceptualization, project administration, B.N.A., S.M.W. and J.M.R.; data curation, methodology, validation, visualization, all authors; formal analysis, writing—original draft, B.N.A.; investigation, S.M.W. and B.N.A.; funding acquisition, supervision, S.M.W. and J.M.R.; writing—review and editing, B.N.A. and J.M.R.; resources, B.N.A. All authors have read and agreed to the published version of the manuscript.

**Funding:** This work was supported by the African Development Bank (AfDB) in partnership with the Ministry of Higher Education Science and Technology and Jomo Kenyatta University of Agriculture and Technology.

**Institutional Review Board Statement:** Not applicable.

**Informed Consent Statement:** Not applicable.

**Data Availability Statement:** The data presented in this study is available upon request from the corresponding author. It is not currently available publicly.

**Conflicts of Interest:** In submitting this manuscript, the authors declare no conflict of interest.

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
