# Peer review of "Abattoir Wastewater Treatment in Anaerobic Co-Digestion with Sugar Press Mud in Batch Reactor for Improved Biogas Yield"

_water, doi:10.3390/w14162571_

Round 1

Author Response

Point 1: Please, be consistent with the use of abbreviations. Once defined, the abbreviation should be used for the entire article. For instance, check mono and co-digestion, slaughterhouse wastewater and sugar press mud through the article.

Response 1: Thank you very much for pointing out that. All the abbreviations have been checked and consistently used in the entire manuscript.

Point 2: Abbreviations such as “it’s” and “didn’t” should never be used in a scientific article. There are random brackets in the manuscript. In some cases, the “0” is used instead of “°” for the “°C” symbol. Please, check the entire manuscript.

Response 2: Thank you very much for the observation. It has been corrected in the entire manuscript.

Point 3: Lines 44 – 46. It seems rather weird that emerging nations consume more meat per person than developed countries. Please, check the accuracy of this info.

Response 3: Thank you very much for the observation. It has been corrected in lines 48-49. 

Point 4: Line 68. What is the difference between lipids and fats? Fats are a type of lipid. Why mention both here?

Response 4: Thank you very much for the observation. This has been corrected in the entire manuscript.

Point 5: Section 2.1. Please, add in this section a table with TS and VS content of the raw substrates, e.g. before dilution to 6%, and inoculum.

Response 5: Thank you very much for the observation. This has been added in section 2.1 of the manuscript.

Point 6: Line 155. 125 mL seems to be rather a small volume for accurate BMP tests. Which was the working volume for this experiment?

Response 6: Thank you very much for the observation. This has been clarified in line 162.

Point 7: Figure 1. The quality of this image is low. Also, the experimental setup is very basic. I believe this picture could be removed, or at least make it more catchy. I would consider using a modified and improved version of this image as graphical abstract more than a figure in the main manuscript.

Response 7: Thank you very much for the observation. Figure 1 has been removed in the main manuscript.

Point 8: There is no need to increase the font size for the formula.

Response 8: Thank you very much for the observation. This has been rectified in the entire manuscript.

Point 9: Section 2.4. Did the authors use the root mean square error (RMSE) to evaluate if the model prediction fits with the experimental data? If that is so, please clarify in the text. Did the authors also perform statistical analysis to assess if the methane production under the various conditions is statistically different? If not, please run the ANOVA test followed by Tucky to compare the cumulated methane productions.

Response 9: Thank you very much for the observation. Yes, the root mean square error (RMSE) was used to evaluate if the model prediction fits with the experimental data using Eq. (2) and the ANOVA test was run under section 2.4 of the manuscript in lines 200-201.  

Point 10: Lines 216-217. Did the authors account for VS reduction due to microorganisms’ growth or death? Please clarify how the VS degradation was calculated.

Response 10: Thank you very much for the observation. The VS reduction was accounted on basis of microorganisms’ growth. This has been clarified in lines 216-217.

Point 11: Lines 253-254. Is 1% sufficient to claim a synergic effect? In any case, please, perform statistical analysis as previously suggested.

Response 11: Thank you very much for the observation. This has been rectified under section 3.2

Point 12: Line 268. Are the TS and VS contents measured after dilution of SPM? Please, clarify. How about SHWW did the authors diluted this sub as well or any other modification before or after storage?

Response 12: Thank you very much for the observation. The TS and VS contents were measured after dilution of SPM. SHWW was not diluted or modified before or after storage. This has been clearly clarified in line 154-155.

Point 13: Lines 276-278. How is it possible that TS and VS content increased by mixing the two substrates? The TS and VS of the mixture should be within the content of the two substrates for both TS and VS. Please, explain this.

Response 13: Thank you very much for the observation. The organic matter content was enriched when SHWW was mixed in different proportions with SPM diluted to 6% TS. This has been corrected in lines 268-269.

Point 14: Lines 293-295. The authors should also discuss here the C/N ratio of SPM. Can the authors report some info regarding the lignocellulosic composition of SPM from the literature?

Response 14: Thank you very much for the observation. The discussion on C/N ratio of SPM and information regarding the lignocellulosic composition of SPM from the literature has been added in lines 302-305.

Point 15: Table 2. How is it possible that the initial tCOD was higher than both 100%SPM and 100% SHWW? If I understood correctly, the initial value of tCOD for the mix conditions should always be between the values of 100% SPM and 100%SHWW. If this is not the case, please clarify this aspect.

Response 15: Thank you very much for the observation. This has been rectified in Table 4.

Point 16: Table 2. Can the authors calculate the C/N ratio also for the mix conditions? This would help to understand how the different mixing influenced this factor. Table 2. Line 298. Does it mean that the first value refers to the mixture, i.e. inoculum + SHWW + SPM, before AD, and the second was measured on the digestate after AD? This is not clear in the manuscript. Please, clarify this in the rebuttal and the entire manuscript.

Response 16: Thank you very much for the observation. The C/N ratios of the mixtures were not tested in this study. In Table 4, yes, the first values refer to the substrates and mixtures before AD and second values in brackets refer to digestate characteristics after AD. It has been made clear in the footnote in lines 286-287.

Point 17: Line 339. What is TPM? TMP maybe?

Response 17: Thank you very much for the observation. TMP stands for theoretical methane potential. This has been corrected in line 405.

Point 18: Line 342. Where does the 65% degradability come from? I cannot see this value in Table2. Please, check it.

Response 18: Thank you very much for the observation. Data on degradability has been deleted in entire manuscript based on comment point “21” of reviewer #1.

Point 19: In the article, the use of biogas or methane is used alternatively referring to the same data (also in the tables and figures). Please, check the entire article carefully. The data should always be reported as mL CH4/g VS. It is okay to report and discuss the CH4 content in biogas, but figures and tables should refer to the specific methane production. Also, the overall mL produced is not of any interest.

Response 19: Thank you very much for the observation. This is well noted and rectified in all Tables, Figures and in the entire manuscript.

Point 20: Table 3 and figure 2. Please, report the standard deviation. Report the specific methane production, i.e. mL CH4/g VS, not mL only. The second line of the table says biogas, but it is methane yield.

Response 20: Thank you very much for the observation. This is well noted and rectified in the Table, Figure and entire manuscript.

Point 21: Table 3. How did the authors calculate the increase in methane yield? This is not clear. Also, what is the unit? Can the authors calculate the BD also for the mixing conditions? If not, this information only for the 100% conditions is not of much interest.

Response 21: Thank you very much for the observation. The synergistic increase in methane yield was calculated according to Eq. (5) in line 241 and it is dimensionless. However, biodegradability (BD) cannot be calculated for mix ratios since it’s calculated based on elemental analysis which was not calculated for the mix ratios. Therefore, the information on BD on 100% substrates has been deleted in the entire manuscript.

Point 22: Lines 284-286. The 40:60 condition does not seem much better as well. Please, perform appropriate statistical analysis as previously suggested.

Response 22: Thank you very much for the observation. This has been rectified in the entire manuscript.

Point 23: Lines 392-395. Please, discuss also the impact of N-NH4 concentration in the other conditions. Check if the standard deviation of this parameter reported in table 2 is correct. It seems too little.

Response 23: Thank you very much for the observation. This has been rectified in Table 4 and in the manuscript in lines 356-359.

Point 24: Line 411. What does his observation refer to?

Response 24: Thank you very much for the observation. This has been revised in lines 374-375.

Point 25: When referring to a mixture of volatile fatty acids, the abbreviation VFAs is commonly used (not VFA).

Response 25: Thank you very much for the observation. This has been rectified in the entire manuscript.

Point 26: Figure 2. The authors reported the methane production in mL, but the unit of the y axis indicates mL/g VS. Please, revised this. The specific methane production should be reported, i.e. mL CH4/g VS.

Response 26: Thank you very much for the observation. This has been rectified in Figure 2 and in the entire manuscript.

Point 27: Line 452. The methane production, i.e. 14 L, should be reported as specific methane production, i.e. i.e. mL CH4/g VS. If the authors are unable to calculate this from the referenced work, please, remove this data (also from Table 4).

Response 27: Thank you very much for the observation. This has been corrected and Table 1has been modified.

Point 28: Table 4. The methane production from this manuscript should be reported as mL CH4/g VS (not mL CH4/g TS).

Response 28: Thank you very much for the observation. This has been corrected and Table 1 has been modified.

Point 29: Lines 462-468. This should be a footnote accompanying Table 4.

Response 29: Thank you very much for the observation. This has been corrected and Table 1 has been modified with the foot note.

Point 30: Table 5. The model should fit the specific methane production, not the mL of methane produced. Please, revise.

Response 30: Thank you very much for the observation. This has been rectified in lines 417-418.

Point 31: Line 492. Why did the authors report two ranges here?

Response 30: Thank you very much for the observation. This has been rectified in line 434.

Point 32: Line 498. Rmax and? Please, check this.

Response 30: Thank you very much for the observation. This has been rectified in line 439.

Reviewer 2 Report

the followings should be observed before making a decision on the manuscript:

1.     Unit of time in page 1 is not mentioned

2.     Abbreviations should be mentioned, between brackets, after full name in the summary.

3.     Titles of tables and figures need to be complete and illustrative.

4.     Rules of hyphenation should be followed allover, including tables.

5.     APHA (2005) is an old edition, the latest is the 2017.

6.     No need for Figure 1

7.     Increase in methane yield “in folds” is missing in Table (3)

8.     Despite the experiment ran for 87 days, results are only given for 66 days without explanation.

9.     It is not clear or supported how ammonia inhibition can be explained in the co-digestion.

10.  C, H, O, and N values should have been included in mixtures (even if calculated) in the relevant tables. 

11.  TS values in Table (2) do not correlate well with either TCOD nor SCOD.

12.  Full analysis of SPM should have been given to help explaining some of the results as it is known that some metals enhance the AD and they are reported to be present in the wastewater of the sugar industry. This would add another important dimension to the discussion.

Author Response

Point 1: Unit of time in page 1 is not mentioned

Response 1: Thank you very much for the observation. This has been rectified.

Point 2: Abbreviations should be mentioned, between brackets, after full name in the summary.

Response 2: Thank you very much for the observation. This has been rectified.

Point 3: Titles of tables and figures need to be complete and illustrative.

Response 3: Thank you very much for the observation. This has been rectified in all Tables and Figure in the entire manuscript.

Point 4: Rules of hyphenation should be followed allover, including tables.

Response 4: Thank you very much for the observation. This has been rectified.

Point 5: APHA (2005) is an old edition, the latest is the 2017.

Response 5: Thank you very much for the observation. This has been rectified in entire manuscript.

Point 6: No need for Figure 1

Response 6: Thank you very much for the observation. The Figure has been deleted in the manuscript.

Point 7: Increase in methane yield “in folds” is missing in Table (3)

Response 7: Thank you very much for the observation. This has been rectified.

Point 8: Despite the experiment ran for 87 days, results are only given for 66 days without explanation.

Response 8: Thank you very much for the observation. This has been rectified.

Point 9: It is not clear or supported how ammonia inhibition can be explained in the co-digestion.

Response 9: Thank you very much for the observation. This has been explained in lines 354-361.

Point 10: C, H, O, and N values should have been included in mixtures (even if calculated) in the relevant tables.

Response 10: Thank you very much for the observation. However, elemental analysis was not tested for the mix proportions.

Point 11: TS values in Table (2) do not correlate well with either TCOD or SCOD.

Response 11: Thank you very much for the observation. This is well noted and rectified.

Point 12: Full analysis of SPM should have been given to help explaining some of the results as it is known that some metals enhance the AD and they are reported to be present in the wastewater of the sugar industry. This would add another important dimension to the discussion.

Response 12: Thank you very much for the observation. However, due to financial limitations the study could not carry out full analysis on SPM apart from those reported.

Reviewer 3 Report

The authors proposed a new strategy for abattoir wastewater treatment with Sugar Pressmud in the Batch reactor for improved Biogas yield. The work is interesting and gives useful insights. The experiments were well conducted, and the results were analyzed and discussed in detail. The manuscript may be accepted in its current form.

Author Response

Point 1: The authors proposed a new strategy for abattoir wastewater treatment with Sugar Press mud in the Batch reactor for improved Biogas yield. The work is interesting and gives useful insights. The experiments were well conducted, and the results were analyzed and discussed in detail. The manuscript may be accepted in its current form.

Response 1: Complements are well appreciated, thank you.

Reviewer 4 Report

The overall research seems interesting and thorough. Besides some minor issues listed below, the main drawback of the paper is that it should be edited to make it more straightforward, as it is hard to follow in the present form. Another drawback are all the formatting issues that need to be corrected.

The more specific issues are listed below:

The formatting of the Equations is bad, the symbols are too big and bold. Please adjust them. It also seems that they don’t all have the same text height.

Line 185 the line break after the word “Where” should be deleted to join this line with the next.

Equation 1 is missing explanations for l and t.

Lines 198 and 199 should be joined, no need for line break after “where”.

Line 225, equations should be referenced as Eq. Or Eqs. Instead of “Equations”.

Furthermore, please make sure to use a uniform referencing style within one Manuscript. Either use Eq. (xx) or Eq. Xx, but always use the same style.

Equation 5 should be altered to use some other notation for the multiplication instead of “X” as it could confuse the reader. There are built-in symbols in the Equation editor in Word.

Tables 2, 3, 4 should be modified to fit on the paper.

Table text formatting should also be synchronized between all the tables within the Manuscript

Figures 2 and 3 should also be modified to have the appropriate font size and style.

Label PRED.SHWW on Fig.3 should have different label as it is displayed as white on the figure and it can’t be seen.

The Authors should include more information regarding the measuring equipment, schematic images, of photos of them, or at least the more important equipment utilized for the implemented research.

Line 287, sentence “The best C/N ratio, according to the literature, is about 20 to”, please include the appropriate reference.

Lines 365 to 379 seem to be more appropriate for the Introduction as they fall in the “Literature review” category.

I would suggest the same for Table 4 as it provides a review of results attained by various Authors, it would be more fitting as a part of the Introduction.

Line 477, the sentence explains the symbol l, this should be done much earlier, when it was used the first time, instead of here.

Author Response

Point 1: The formatting of the Equations is bad, the symbols are too big and bold. Please adjust them. It also seems that they don’t all have the same text height.

Response 1: Thank you very much for the observation. This has been rectified in entire manuscript.

Point 2: Line 185 the line break after the word “Where” should be deleted to join this line with the next.

Response 2: Thank you very much for the observation. This has been rectified.

Point 3: Equation 1 is missing explanations for l and t.

Response 3: Thank you very much for the observation. This has been rectified.

Point 4: Lines 198 and 199 should be joined, no need for line break after “where”.

Response 4: Thank you very much for the observation. This has been rectified.

Point 5: Line 225, equations should be referenced as Eq. Or Eqs. Instead of “Equations”. Furthermore, please make sure to use a uniform referencing style within one Manuscript. Either use Eq. (xx) or Eq. Xx, but always use the same style.

Response 5: Thank you very much for the observation. This has been rectified.

Point 6: Equation 5 should be altered to use some other notation for the multiplication instead of “X” as it could confuse the reader. There are built-in symbols in the Equation editor in Word.

Response 6: Thank you very much for the observation. This has been rectified.

Point 7: Tables 2, 3, 4 should be modified to fit on the paper.

Response 7: Thank you very much for the observation. This has been rectified.

Point 8: Table text formatting should also be synchronized between all the tables within the Manuscript. Figures 2 and 3 should also be modified to have the appropriate font size and style.

Response 8: Thank you very much for the observation. This has been rectified.

Point 9: Label PRED.SHWW on Fig.3 should have different label as it is displayed as white on the figure and it can’t be seen.

Response 9: Thank you very much for the observation. This has been rectified.

Point 10: The Authors should include more information regarding the measuring equipment, schematic images, of photos of them, or at least the more important equipment utilized for the implemented research.

Response 10: Thank you very much for the observation. This has been added in section 2.2.

Point 11: Line 287, sentence “The best C/N ratio, according to the literature, is about 20 to”, and please include the appropriate reference.

Response 11: Thank you very much for the observation. This has been rectified in lines 277-278.

Point 12: Lines 365 to 379 seem to be more appropriate for the Introduction as they fall in the “Literature review” category.

Response 12: Thank you very much for the observation. This has been rearranged and moved to introduction section.

Point 13: I would suggest the same for Table 4 as it provides a review of results attained by various Authors, it would be more fitting as a part of the Introduction.

Response 13: Thank you very much for the observation. This has been modified and moved to introduction section as Table 1.

Point 14: Line 477, the sentence explains the symbol l, this should be done much earlier, when it was used the first time, instead of here.

Response 14: Thank you very much for the observation. This has been rectified in line 427.

Round 2

Reviewer 2 Report

Most comments were addresses properly, however there are still some needed modifications:

1. the introduction (Table 1) contains results (present study) it should be moved to the results section.

2. There is no need for Fig. 1, it does not add to the manuscript.  

Author Response

Point 1: The introduction (Table 1) contains results (present study) it should be moved to the results section.

Response 1: The  data on present study has been omitted in Table 1 since its presented in results and discussion section of the manuscript.

Point 2: There is no need for Fig. 1, it does not add to the manuscript. 

Response 2: Figure 1 has been omitted in the entire manuscript.

This manuscript is a resubmission of an earlier submission. The following is a list of the peer review reports and author responses from that submission.